# Altered Resting-State Brain Activity and Functional Connectivity in Post-Stroke Apathy: An fMRI Study

**DOI:** 10.3390/brainsci13050730

**Published:** 2023-04-27

**Authors:** Shiyi Jiang, Hui Zhang, Yirong Fang, Dawei Yin, Yiran Dong, Xian Chao, Xiuqun Gong, Jinjing Wang, Wen Sun

**Affiliations:** 1Stroke Center & Department of Neurology, The First Affiliated Hospital of USTC, Division of Life Sciences and Medicine, University of Science and Technology of China, Hefei 230001, China; jiangshiyi@mail.ustc.edu.cn (S.J.); 13956253980@163.com (Y.F.); dongyiran@mail.ustc.edu.cn (Y.D.); chaoxian0828@mail.ustc.edu.cn (X.C.); 2Department of Gastroenterology, Zhongshan Hospital of Traditional Chinese Medicine, Zhongshan 528400, China; 13822718962@163.com; 3Department of Radiology, The First Affiliated Hospital of USTC, Division of Life Sciences and Medicine, University of Science and Technology of China, Hefei 230001, China; david2855952023@163.com; 4Department of Neurology, The First Affiliated Hospital of Anhui University of Science and Technology, Huainan First People’s Hospital, Huainan 232000, China; 5Department of Neurology, Nanjing Jinling Hospital, Affiliated Hospital of Medical School, Nanjing University, Nanjing 210033, China

**Keywords:** resting-state fMRI, ischemic stroke, post-stroke apathy, fractional amplitude of low-frequency fluctuation, functional connectivity

## Abstract

Apathy is a common neuropsychiatric disease after stroke and is linked to a lower quality of life while undergoing rehabilitation. However, it is still unknown what are the underlying neural mechanisms of apathy. This research aimed to explore differences in the cerebral activity and functional connectivity (FC) of subjects with post-stroke apathy and those without it. A total of 59 individuals with acute ischemic stroke and 29 healthy subjects with similar age, sex, and education were recruited. The Apathy Evaluation Scale (AES) was used to evaluate apathy at 3 months after stroke. Patients were split into two groups—PSA (*n* = 21) and nPSA (*n* = 38)—based on their diagnosis. The fractional amplitude of low-frequency fluctuation (fALFF) was used to measure cerebral activity, as well as region-of-interest to region-of-interest analysis to examine functional connectivity among apathy-related regions. Pearson correlation analysis between fALFF values and apathy severity was performed in this research. The values of fALFF in the left middle temporal regions, right anterior and middle cingulate regions, middle frontal region, and cuneus region differed significantly among groups. Pearson correlation analysis showed that the fALFF values in the left middle temporal region (*p* < 0.001, r = 0.66) and right cuneus (*p* < 0.001, r = 0.48) were positively correlated with AES scores in stroke patients, while fALFF values in the right anterior cingulate (*p* < 0.001, r = −0.61), right middle frontal gyrus (*p* < 0.001, r = −0.49), and middle cingulate gyrus (*p* = 0.04, r = −0.27) were negatively correlated with AES scores in stroke patients. These regions formed an apathy-related subnetwork, and functional connectivity analysis unveiled that altered connectivity was linked to PSA (*p* < 0.05). This research found that abnormalities in brain activity and FC in the left middle temporal region, right middle frontal region, right cuneate region, and right anterior and middle cingulate regions in stroke patients were associated with PSA, revealing a possible neural mechanism and providing new clues for the diagnosis and treatment of PSA.

## 1. Introduction

Post-stroke apathy (PSA) is a common neuropsychiatric disease after stroke [1], described by diminished goal-directed behavior of cognitive, emotional, social domains [2]. PSA may occur in all phases of stroke, including the acute, subacute, and chronic phases, with an estimated prevalence of approximately 1/3 [3]. Previous studies have shown that apathy hinders recovery from stroke, affects quality of life [4], leads to cognitive impairment [5], and causes death in stroke patients [6]. Additionally, it is a huge burden and distress for the patient’s family and caregivers [7].

The pathogenesis of post-stroke apathy is unclear and may be related to a variety of factors, such as the site, extent, and severity of brain injury, as well as the patient’s age, gender, education, social support, cognitive function, and depressive symptoms [8]. Apathy following stroke, according to some research, is linked to damage to the gyrus, such as the anterior cingulate, Roland synaptic area, and basal ganglia; they are closely associated with emotion regulation and reward systems [9,10,11]. It is currently believed that only the location of a lesion does not sufficiently reflect that distant structural or functional changes contribute to apathy, and even whether there is an connection between the location of a lesion and post-stroke apathy [12]. PSA is accompanied by focal lesions in key network areas or disruption of intranetwork connections [13]. Apathy was associated with the limbic system, frontal, temporal, etc. [14]. It is noteworthy that imaging data were obtained within 7 days after stroke onset, whereas the assessment of apathy was performed within 1 month after stroke. Therefore, a conclusion can be drawn that the beginning of apathy 1 month later may be predicted by early alterations in the structural brain network connectivity between these areas.

The blood-oxygen-level-dependent (BLOD) signal that reflects task-independent nondirectional brain activity is the foundation of resting-state functional magnetic resonance imaging (rs-fMRI), and rs-fMRI is popularly used in stroke populations [15,16]. The amplitude of low-frequency fluctuation (ALFF) gives an exact description in spontaneous brain activity for each voxel and is a popular way to characterize the local neural activity of the brain [17]. The interference of physiological noise, however, raises the uncertainty of study findings in actual uses. An improved fractional amplitude of low-frequency fluctuation (fALFF) developed by a low-frequency amplitude can deal with this problem and eliminate signal artifacts of the non specific brain neuron activity [18]. Functional connectivity (FC) shows a relationship between the spontaneous activity of functionally related brain regions in time series, reflecting how functional activity and different brain regions are synchronized. Meanwhile, research has revealed that brain regions are made up of specific networks [13]. In patients with parenchymal injury, the volume and location of a lesion may be less susceptible than an abnormal spontaneous local nerve activity of the brain [19].

Recently, researchers have been able to examine the altered functional brain connectivity of various neuropsychiatric disorders due to rs-fMRI [20,21]. In previous studies, it was discovered that abnormal fALFF in the prefrontal region and precentral region were linked to poststroke depression, the fALFF of the left insula was linked to the depression rigidity as well [19]. In addition, the alterations in neural activity and FC in cognition-related regions detected by rs-fMRI are good indicators to elucidate the mechanism of action of repetitive transcranial magnetic stimulation on cognitive dysfunction after stroke [22]. However, no research has yet examined the association of ischemic stroke individuals with apathy, brain activity, and FC. As a result, the current study was carried out to address this issue.

This research sought to discover the interaction between functional cerebral activity and apathy severity of individuals with post-stroke apathy, as well as changes in functional connectivity at these important nodes. Our study provides new perspectives and evidence to uncover the neural mechanisms of post-stroke apathy and to assess its diagnosis and prognosis. The following is how this essay is organized: Section 1 introduces the background of post-stroke apathy; Section 2 describes our study methods, including the recruitment and screening of participants, data collection and processing, and the steps and methods of statistical analysis; Section 3 reports our findings, including the variation of fALFF values and functional connectivity and the correlation between fALFF and apathy severity; Section 4 discusses the significance and limitations of this research and directions for future research; Section 5 summarizes this paper. This paper proposes the following hypotheses: (1) there are significant neuroimaging differences between post-stroke apathy patients and non post-stroke apathy patients; (2) damage to these regions causes abnormal functional connectivity in PSA. We anticipate that this study can improve the current understanding of PSA neuropathology through rs-fMRI.

## 2. Materials and Methods

The materials and methods used by this research would be described in this chapter. This includes the subjects who participated in the study, as well as any tools that were used to collect data and statistical methods that were used to study. In Section 2.1, we will provide more detailed information about the subjects and their characteristics.

### 2.1. Participants

Acute ischemic stroke patients admitted by the First Affiliated Hospital of the University of Science and Technology of China between March 2021 and December 2021 were prospectively screened for the research. 

The admission and exclusion criteria of stroke patients were followed. The admission criteria (1) were 18 years or older, (2) were right-handed, (3) signed a written informed consent, and (4) were diagnosed with first acute ischemic stroke within 2 weeks of stroke onset. The exclusion criteria were (1) combination with other neurological diseases that are prone to apathy; (2) history of apathy, depression, and other mental illness; (3) contraindication to MRI (metal or electronic device implants, cranial defects, etc.); (4) aphasia or comprehension impairment; and (5) minimum mental state examination (MMSE) < 17 or inability to complete the scale. 

The healthy controls were all included if they fit the following requirements: (1) 18 years or older and (2) right-handed. Healthy controls were excluded in the case of (1) previous history of cerebrovascular disease; (2) history of apathy, depression, and other psychiatric disorders; (3) severe aphasia or comprehension impairment; (4) contraindications to MRI (metal or electronic device implants, cranial defects, etc.); and (5) MMSE scores < 17 or inability to complete the scale.

In addition, if image quality problems occurred during the data preprocessing phase, the subjects were also excluded. Additionally, if image quality problems, such as severe motion artifacts, distortion, or low signal-to-noise ratio, occurred during the data preprocessing phase, the subjects were also excluded. A total of 11 subjects (7 stroke patients and 4 healthy controls) were excluded due to image quality problems. The final sample consisted of 59 stroke individuals and 29 healthy controls.

This research was approved by the Ethics Committee of the First Affiliated Hospital of the University of Science and Technology of China, and written informed consent was obtained from all participants.

### 2.2. Clinical and Neuropsychological Evaluations

Apathy was assessed 3 months after stroke, as other studies have verified [23]. It was diagnosed on stroke patients using criteria of brain disorders for apathy. All patients had to present with one or both of the main symptoms (loss of interest or pleasure, low social initiative, etc.) lasting more than 4 weeks. The seriousness of apathy was evaluated by the Apathy Evaluation Scale (AES). It is considered a dimension of apparent behavioral, cognitive, and emotional deficits and consists of 18 items, whose score is in the range of 18 and 72; high scores indicate strong apathy. The reliability and internal consistency of AES was previously confirmed [24]. This scale was initially validated in stroke patients [25] and Asian patients [14]. The Hamilton Depression Scale (HAMD)-24, Lubben Social Network Scale (LSNS), and MMSE scale were used to assess depression, social communications, and cognitive impairment, respectively.

Baseline clinical characteristics were collected from all patients at admission. The National Institutes of Health Stroke Scale (NIHSS) was used to measure stroke seriousness, and the Modified Rankin Scale (mRS) to estimate each patient’s level of functional disability.

### 2.3. Imaging Data Acquisition

A 3.0 T MRI Scanner (GE, Boston, MA, USA) with a 24-Channel Receiver Array Head Coil Scanner was used to collect imaging data. After instructing all subjects to close their eyes with sponge earplugs worn and remain awake all the time, an 8 min rs-fMRI scan (TR = 2000 ms; TE = 30 ms; flip angle = 90; slices = 36; thickness= 3 mm; gap = 1 mm; matrix 64 × 64; FOV = 200 × 200 mm) was performed. After functional scanning, the T1-weighted images of the whole brain (TR = 8 ms; TE = 3 ms; flip angle = 12; thickness = 1 mm; gap = 1 mm; FOV = 100 × 100 mm) and diffusion-weighted images were measured.

### 2.4. Preprocessing

A Resting-State fMRI Data Analysis Toolkit plus (restplus) toolbox (http://www.restfmri.net/forum/, V1.27, Institute of Psychology, Chinese Academy of Sciences, Beijing, BJ, China, accessed on 1 April 2022) [26] was used to perform data preprocessing. Preprocessing includes the following steps: (1) the first 10 volumes were removed to account for impacts of scanning and the subject’s prestability, (2) all slices’ scanning time must be aligned to the reference slice by correcting the time layer, (3) head motion correction was performed to remove subjects with head translation over 1.5 mm and rotation over 1.5 degrees, (4) adjusting to MNI coordinates was made, and the size of voxels was changed to 3×3×3 mm^2^, (5) a spatial filter with a 6 mm Gaussian curve at half its maximum height was applied to minimize the impact of incomplete alignment and improve signal-to-noise ratio, (6) detrending was performed to remove linear drift from the spatially smoothed images; and (7) regression of interference parameters was performed to correct for head motion and physiological noise. Filtering is required when calculating functional connections.

### 2.5. fALFF Calculation

A square derivative whose frequency ranges from 0.01 to 0.08 Hertz was averaged to calculate the fALFF to lower the responsiveness of the ALFF of physiological sound. The sensitivity and specificity of local cerebral activity detection can be considerably improved by this spectrum, which can effectively suppress nonspecific signal components in rs-fMRI [27]. The average fALFF was used to divide each voxel’s fALFF for standardization purposes, which reduces the effect of individual subject variability and being a useful tool for assessment. This index has been widely used to study local neurological function in neurological and psychiatric disorders [19,28].

### 2.6. Infarct Volume and Surface Thickness

The lesions were tracked semiautomatically on DWI images using ITK-SNAP [29] software (http://www.itksnap.org/pmwiki/pmwiki.php, V3.4.0, Penn Image Computing and Science Laboratory, Philadelphia, PA, USA, accessed on 15 April 2022). The tracked images were visually inspected and validated by a stroke neurologist so that infarct volumes were automatically evaluated. The computational anatomy toolbox (CAT12: http://www.neuro.uni-jena.de/cat/, Friedrich-Schiller-Universitt Jena, Jena, Germany, accessed on 15 April 2022) for SPM (Statistical Parametric Mapping software, http://www.fil.ion.ucl.ac.uk/spm/, Wellcome Department of Cognitive Neurology, University College London, London, UK, accessed on 15 April 2022) was used on T1 images to calculate surface thickness and total intracranial volume (TIV).

### 2.7. Statistical Analysis

The demographic, neuropsychological, and neuroimaging variables were compared using one-way ANOVA among the PSA, nPSA, and HCs groups. ANOVA is a common statistical method that can be used to compare differences between groups [30,31,32,33]. The Statistical Package for the Social Sciences (SPSS26.0) was used to analyze our data.

The two-sample *t*-test was used to measure whether there may be any variations between the PSA and nPSA groups. Nonparametric data were analyzed using the Mann–Whitney–Wilcoxon test, and categorical data were analyzed using the chi-square test or Fisher’s exact test. A 0.05 significant-level two-tailed analysis was used to report the probability levels.

The fALFF values were calculated using the Restplus toolbox, and one-way ANOVA was used to generate F-plots with Gaussian random field (GRF) correction in the PSA, nPSA, and HCs groups and the correction’s voxel-level p<0.05 and cluster-level p<0.05, and it was two-tailed. Based on above findings, a post hoc analysis with GRF correction to confirm the differences between the PSA and nPSA groups was performed, as well as the PSA and HCs groups and the nPSA and HCs groups and the correction’s voxel-level p<0.01 and cluster-level p<0.01, and it was two-tailed. The NIHSS scores, age, sex, and education were the covariates of the preceding analysis. The relationship of the fALFF values and AES scores was analyzed by Pearson correlation. Based on the significant fALFF regions, we defined regions of interest (ROIs) for further analysis. Before statistical analysis, the FC maps between ROIs were calculated and z-transformed. The FC values of the ROIs were compared between groups using the two-sample *t*-test to see if there were any variations (PSA vs. nPSA groups, PSA vs. HCs groups, nPSA vs. HCs groups).The BrainNet Viewer toolkit was used to display the results [34].

## 3. Results

The outcomes of this research will be laid out in this chapter, which includes findings from our analysis of the data collected from the subjects. In Section 3.1, we will provide more detailed information about the demographic characteristics of the subjects and how they relate to our findings.

### 3.1. Demographic Data

Sex, age, education level, HAMD score, smoking, BMI, onset to imaging time, and stroke circulation were not significantly different among subjects with PSA, nPSA, and HCs, in accordance with Table 1. The MMSE, AES, and HAMD ratings differed significantly between the groups. The PSA group outscored the nPSA group in terms of NIHSS, discharged mRS, and AES scores, but came low in terms of MMSE scores. TIV and surface thickness revealed no differences. 

### 3.2. Post-Stroke Apathy and fALFF

We found significant differences in the frontal cortex, cingulate cortex, and temporal cortex brain regions in the PSA, nPSA, and HCs groups by using one-way ANOVA and GRF correction (p<0.05 at the voxel level and p<0.05 at the cluster level, GRF correction) (Appendix A). A post hoc two-sample *t*-test was applied to further explore brain regions where the significant differences arose (voxel-level p<0.01 and cluster-level p<0.01, GRF correction); it showed that the PSA group had higher fALFF values in the left middle temporal region and lower fALFF values in the right anterior cingulate region and right middle frontal region compared with the nPSA group (Figure 1A). Compared with the HCs group, the PSA group had higher fALFF values in the left middle temporal region and right cuneate region and lower fALFF values in the right middle cingulate region and frontal middle region (Figure 1B). However, the nPSA and HCs groups did not have any substantially different regions. Table 2 shows the peak coordinates of the Montreal Neurological Institute. The T-score indicates how different the two groups are from one another in units of standard deviation. A larger t-value indicates a more significant difference. X, Y, and Z represent the dimensions of the brain region.

Based on the above significant clusters, the fALFF values in the patients from the three groups from the results for further analysis were extracted (Table 2). The fALFF values of the right cuneus brain region and the left middle temporal brain region were both significantly increased in the PSA group compared with nPSA group. Moreover, the PSA group showed much decreased fALFF in the right cingulate region and middle frontal region compared with the nPSA group (Figure 2).

The fALFF values in the left middle temporal region (*p* < 0.001, r = 0.66) (Figure 3A) and right cuneus (p<0.01, r=0.48) (Figure 3C) and fALFF values in the right anterior cingulate brain regions (p<0.001, r=−0.61) (Figure 3B), right middle frontal region (p<0.001, r=−0.49) (Figure 3C), and right middle cingulate region (p=0.04, r=−0.27) (Figure 3E) were negatively correlated with AES scores in stroke patients, according to Pearson correlation analysis.

### 3.3. Post-Stroke Apathy and Local Functional Connectivity

This research further selected five ROIs (peak MNI coordinates are listed in Table 2) for the next step of functional connectivity analysis. Compared with the nPSA group, FCs in the PSA group between the left middle temporal region and right cuneus, right middle frontal region and right cuneus, and right middle frontal region and right middle cingulate region were decreased. The FCs between the left middle temporal region and right anterior cingulate region, left middle temporal region and right middle cingulate region, left middle temporal region and right middle frontal region, right anterior cingulate region and right middle cingulate region, right anterior cingulate region and right cuneus, right middle frontal region and right anterior cingulate region, and right middle cingulate region and right cuneus were increased in the PSA group (Figure 4). Similarly, all FCs in the five regions were increased in the PSA group compared with the HCs group.

## 4. Discussion

This research was the initial effort to elaborate on how PSA and spontaneous brain activity are related, as far as the authors are aware. This study indicated that fALFF in the left middle temporal region, right middle frontal region, right cuneus, right anterior, and middle cingulate region were altered in PSA patients. This alteration significantly correlated with the severity of PSA. Furthermore, the altered functional connectivity of these important regions may be one of the potential neural mechanisms of PSA. 

This study’s primary goals were to pinpoint the regions of the brain where fALFF showed local neural activity and to determine whether functional connectivity was related to lethargy in stroke patients. Consistent with prior studies [35], it was found that abnormal spontaneous neural activity in the cingulate [36], frontal cortex [37], temporal lobe [38], and cuneus [39] was associated with apathy. The cingulate correlates with emotional response and emotional behavior, and there is evidence [40] that damage to the cingulate region can lead to apathy to pain and other emotion-related sensations and that damage there can also lead to social apathy and emotional apathy [41]. Moreover, apathy is significantly associated with frontal lobe activity, emotional apathy is the most common behavioral change related to prefrontal or basal ganglia damage, and its occurrence is related to prefrontal and basal ganglia dysfunctional loops [42]. In extensive temporal lobe lesions or bilateral temporal lobe lesions [43], the most common symptoms are personality changes, abnormal mood, memory impairment, mental retardation, and apathy.

In addition, these regions were found to be important nodes involved in large-scale intrinsic connectivity networks (ICNs), such as the default mode network (DMN), central executive network (CEN), salience network (SN), and limbic system. The cingulate region is part of the limbic and SN systems and an important region in the CEN and DMN. Specifically, the DMN is associated with social cognition, executive function, and other functions [44]. When the brain is in a resting state involved in concentration and in a resting state, there is structural and functional interconnection of internal processing in the DMN [45], such as introspective thinking and imagination. While the brain is resting and not actively engaged in focused, goal-oriented tasks, the “default mode” or subconscious activity of the brain increases for processing internal thought, most likely evolving from an evolutionary perspective of self-preservation [46]. CEN is responsible for task and decision-making tasks and processes a range of different information, such as flexibility, working memory, priming, and inhibition, functions that were previously considered independent processing [16,47]. The functions of the SN involve cognition, emotion, and motivation, and it can regulate the two main control networks of the brain, the DMN and CEN, and realize the conversion of internal and external processing [48]. The limbic system regulates many core brain functions, including response, reaction, behavior, emotion, memory, and learning [49]. Additionally, it is found that the right cuneus has higher fALFF, which is highly favorably associated with apathy. Previous research has connected the cuneus, which is a component of the DMN, to abnormal affective and interoceptive processing [50]. It has been repeatedly observed that the cuneus plays a central role in recalling personal past events [51], which is called autobiographical memory, performed significantly in apathy [52]. The frontal gyrus, as part of the DMN, is strongly associated with apathy, which is consistent with previous studies [42,53,54]. The middle temporal gyrus is an important node of the CEN. As mentioned earlier, the CEN seems to play an important role in apathy [55]. It controls social, emotion and motivation domains, which constitute the core of apathy [56]. In summation, apathy, a complicated disease described by a lack of activation, interest, and emotional reactivity, and PSA might happen because of different areas and changed links in the DMN, CEN, SN, and emotional system. 

According to this research, there were no differences in stroke circulation, infarct volume, cortical thickness, or TIV volume in the PSA and nPSA groups. These results are consistent with Brodaty’s study and van Dalen’s systematic review [56,57]. Furthermore, the notable grouping between the nPSA and HCs groups was not discovered, which might be attributed to stroke severity and small sample size; however, to fully understand this in the future, more research is required.

Furthermore, in line with prior research [58,59,60], it is shown that the PSA group tended to have lower levels of cognition, more severe strokes, and poorer recovery than healthy controls in our study. The incompetence in goal-direction thinking and behavior can be used to explain the connection between PSA and cognitive impairment, which could result in cognitive testing participants losing interest and not trying with effort [61,62]. Stroke severity contributes to apathy to some extent, and it causes or exacerbates the changes in apathy [63]. PSA is linked to functional disability consisting of reduced daily activities and poorer functional recovery [56]. The factors behind these relationships have not been explored [64], although the loss of self-motivation can be speculated to be implicated in reducing engagement in rehabilitation programs [63]. HAMD scores differed between the PSA, nPSA, and HCs groups and did not differ between the PSA and nPSA groups, but there was a significant trend. The lack of significant difference may be due to the small amount of data.

Additionally, our study has some limitations. First, this study’s sample size was relatively small. Second, only 3 months after stroke was PSA assessed, and no long-term dynamic follow-up was provided; thus, the analyzation of the correlation between the dynamic changes in PSA and image is difficult in the features. Third, the NIHSS scores of the patients were relatively low and not representative of all stroke populations. Finally, this study is only based on rs-fMRI and does not involve the DTI structural network. To more visually demonstrate the associations and differences between our study and similar studies, we summarize our main results and findings in Table 3 and compare them with other studies.

## 5. Conclusions

Apathy is a complicated disorder described by a lack of activation, interest, and emotional reactivity. In this study, it was found that fALFF in the left middle temporal region, right anterior and middle cingulate regions, right middle frontal region, and right cuneus region was altered and correlated with apathy seriousness, which are impaired of apathy. Moreover, we found that functional connectivity consisting of some important nodes in the DMN, CEN, SN, and limbic system was also altered in the PSA group. These networks are responsible for self-referential processing, executive functioning, salience detection, and emotional processing, which are also affected by apathy. Apathy may be linked to abnormal activity and connectivity of multiple cerebral regions and networks that underlie key cognitive and affective processes, according to our results. This is the first research to look into both fALFF and functional connectivity changes in PSA. The changes we observed may aid in the differential diagnosis and disease progression monitoring of PSA. Large prospective multicenter cohorts with longitudinal follow-up and comprehensive assessments are required to confirm and extend our findings.

## Figures and Tables

**Figure 1 brainsci-13-00730-f001:**
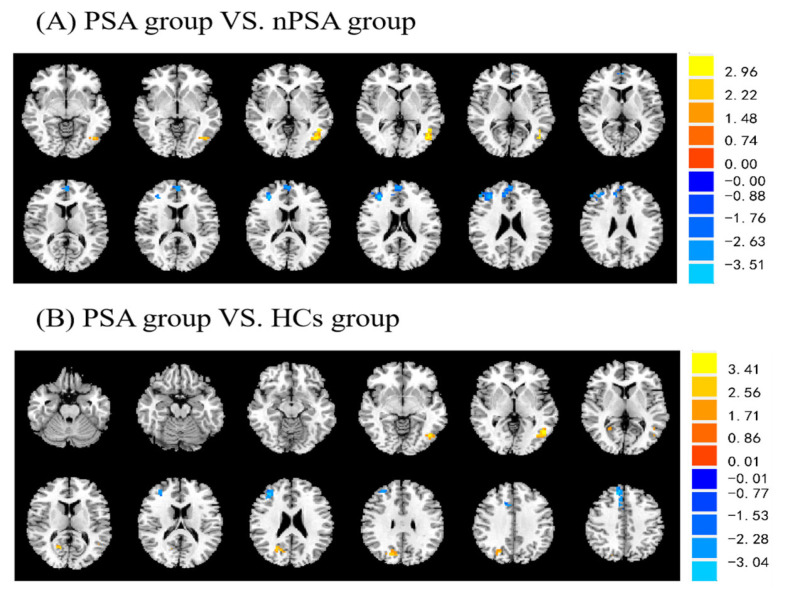
Differences of fALFF between the PSA group and the nPSA group (**A**) and the PSA group and the HCs group (**B**). Voxel-level p<0.01, cluster-level p<0.01, GRF correction. Regions with greater fALFF are represented by yellow areas, while those with lesser fALFF are represented by blue areas. PSA: post-stroke apathy; nPSA: non-post-stroke apathy; HCs: healthy controls; fALFF: fractional amplitude of low-frequency fluctuation; GRF correction: Gaussian random field correction. Different colors indicate high or low T-values.

**Figure 2 brainsci-13-00730-f002:**
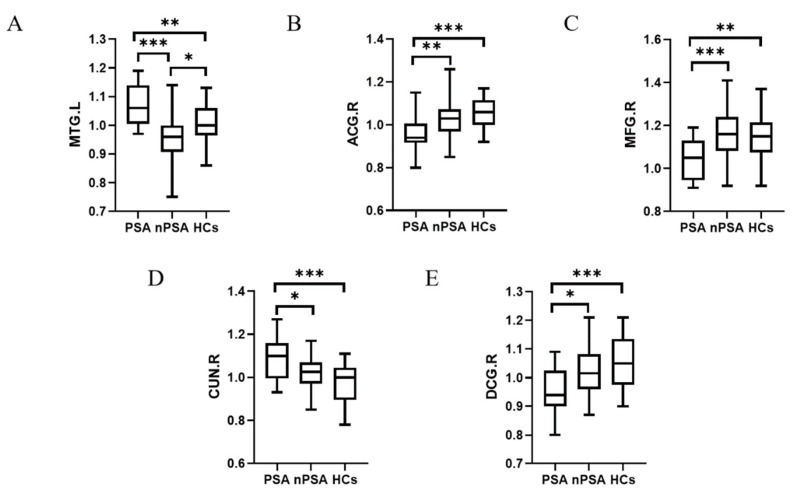
Comparison of mean fALFF values in PSA, nPSA, HCs groups. (**A**) fALFF values of three groups in MTG.L; (**B**) fALFF values of three groups in ACG.R; (**C**) fALFF values of three groups in MFG.R; (**D**) fALFF values of three groups in CUN.R; (**E**) fALFF values of three groups in DCG.R. PSA: post-stroke apathy; nPSA: non-post-stroke apathy; HCs: healthy controls; ACG: anterior cingulate region; MTG: temporal_Mid_L; DCG: median cingulate region; MFG: middle frontal gyrus; CUN: cuneus; *: p<0.05; **: p<0.005; ***: p<0.0005.

**Figure 3 brainsci-13-00730-f003:**
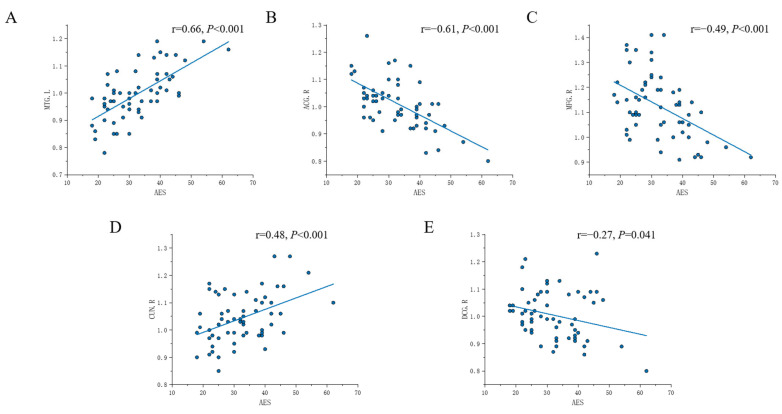
Scatter plots displaying the correlations of fALFF values and the AES scores in stroke patients. (**A**) The correlations of the fALFF values in MTG.L and the AES scores; (**B**) the correlations of the fALFF values in ACG.R and the AES scores; (**C**) the correlations of the fALFF values in MFG.R and the AES scores; (**D**) the correlations of the fALFF values in CUN.R and the AES scores; (**E**) the correlations of the fALFF values in DCG.R and the AES scores. fALFF: fractional amplitude of low-frequency fluctuation; AES: Apathy Evaluation Scale; MTG: middle temporal gyrus; ACG: anterior cingulate region; MFG: middle frontal gyrus; CUN: cuneus; DCG: median cingulate region.

**Figure 4 brainsci-13-00730-f004:**
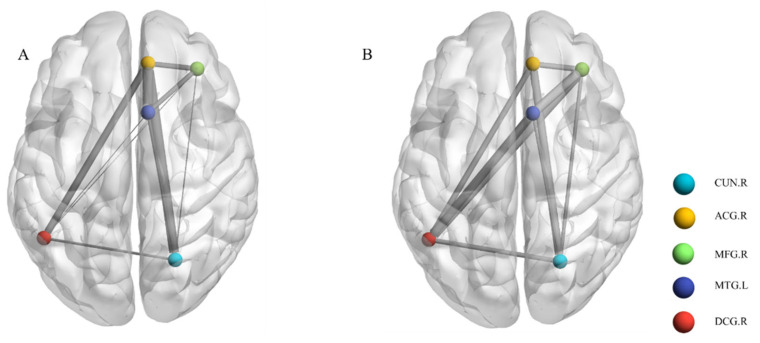
Local functional connectivity subnetwork connections between the two groups. (**A**) PSA group and nPSA group; (**B**) PSA group and HCs group. Age, sex, education and NIHSS scores were used as covariates, *p* < 0.05. Blue, node in the right cuneus; orange, node in the right anterior cingulate region; green, node in the right middle frontal region; navy blue, node in the left middle temporal region; red, node in the right middle cingulate region. PSA: post-stroke apathy; nPSA: non-post-stroke apathy; HCs: healthy controls; MTG: Temporal_Mid_L; ACG: anterior cingulate region; MFG: middle frontal region; CUN: cuneus; DCG: median cingulate region.

**Table 1 brainsci-13-00730-t001:** Demographic and clinical characteristics of stroke patients and healthy controls.

	PSA (*n* = 21)	nPSA (*n* = 38)	HCs (*n* = 29)	*P* _1_	*P* _2_
Age, mean (SD), years	61.7 (11.8)	58.4 (12.5)	56.2 (8.9)	0.33	0.32
Female, *n* (%)	9 (42.9)	12 (31.6)	12 (41.4)	0.43	0.60
Education < 12, *n* (%)	15 (71.4)	23 (60.5)	20 (69.0)	0.40	0.64
BMI, mean (SD), kg/m^2^	24.1 (4.04)	23.4 (3.7)	24.4 (2.0)	0.28	0.38
Smoking, *n* (%)	6 (28.6)	15 (39.5)	6 (20.1)	0.40	0.25
MMSE, mean (SD)	21.9 (3.1)	23.8 (2.7)	25.46 (2.1)	0.02	<0.001
AES, median (IQR)	41.2 (38.9, 45.2)	26.0 (22.8, 31.1)	23.5 (21.7, 27.4)	<0.001	<0.001
HAMD, median (IQR)	4.7 (1.6, 8.0)	3.0 (1.6, 7.2)	1.1 (0.2, 7.5)	0.06	<0.001
LSNS, median (IQR)	26.0 (22.3, 28.8)	29.0 (25.0, 34.0)	30.0 (23.0, 34.0)	0.06	0.07
Imaging time, mean (SD)	13.8 (1.1)	15.4 (2.4)	-	0.43	-
Infarct volumes, cm^3^ (SD)	6.5 (2.3)	4.6 (1.6)	-	0.32	-
NIHSS median (IQR)	6.0 (2.4, 8.4)	0.6 (2.5, 5.3)	-	0.02	-
mRS, median (IQR)	1.2 (1.0, 2.0)	1.0 (0.8, 1.3)	-	0.04	-
TIV cm^3^, mean (SD)	1450 (146)	1304 (137)	1355 (121)	0.36	0.56
Surface thickness mm (SD)	2.16 (0.60)	2.26 (0.57)	2.20 (0.64)	0.53	0.64
Anterior circulation, *n* (%)	15 (71.4)	24 (63.2)	-	0.45	-
Posterior circulation, *n* (%)	6 (28.6)	8 (21.1)	-	0.71	-

PSA: post-stroke apathy; nPSA: non-post-stroke apathy; HCs: healthy controls; LSNS: Lubben Social Network Scale; HAMD: Hamilton Depression Scale; AES: Apathy Evaluation Scale; mRS: Modified Rankin Scale; NIHSS: National Institutes of Health Stroke Scale; MMSE: Mini-Mental State Examination. Imaging time indicates the time from stroke onset to MRI scanning; *P*_1_ indicates differences between the PSA and nPSA groups; *P*_2_ indicates differences among the PSA, nPSA, and HCs groups.

**Table 2 brainsci-13-00730-t002:** Brain regions showing different fALFF values between every two groups.

	Peak Location MNI (mm)			T-Score	Cluster Size
	X	Y	Z		
**PSA versus nPSA**					
Temporal_Mid_L	−48	−54	3	3.70	38
Cingulum_Ant_R	9	42	24	−3.43	13
Frontal_Mid_R	36	39	21	−4.03	72
**PSA versus HCs**					
Temporal_Mid_L	−48	−63	0	4.26	28
Cuneus_R	24	−66	21	3.31	33
Frontal_Mid_R	36	39	24	−3.50	59
Cingulum_Mid_R	9	15	39	−3.51	22

PSA: post-stroke apathy; nPSA: non-post-stroke apathy; HCs: healthy controls; fALFF: fractional amplitude of low-frequency fluctuation; Temporal_Mid_L: middle temporal region; Frontal_Mid_R: middle frontal region; Cingulum_Mid_R: median cingulate region. MNI: Montreal Neurological Institute; Cingulum_Ant_R: anterior cingulate region.

**Table 3 brainsci-13-00730-t003:** Comparison of this study with similar studies.

Title	Year	Cohort Size	Main Achievements and Results
Altered resting-state brain activity and functional connectivity in post-stroke apathy: an fMRI study	2023	PSA (*n* = 21) nPSA (*n* = 38) HCs (*n* = 29)	Abnormalities in brain activity and functional connectivity in the left middle temporal region, right middle frontal region, right cuneate gyrus, right anterior, and middle cingulate region were associated with PSA, revealing a possible neural mechanism and providing new clues for the diagnosis and treatment of PSA.
Resting-state fMRI analysis in apathetic Alzheimer’s disease [65]	2020	aAD (*n* = 10) naAD (*n* = 10) CN (*n* = 10)	Motivation to start acting seems to be controlled by regions near and inside the pregenual anterior cingulate cortex, and this function seems to weaken when apathy gets worse in AD.
Abnormal brain activities in multiple frequency bands in Parkinson’s disease with apathy [66]	2022	PD-A (*n* = 28) nPD-A (*n* = 19) HCs (*n* = 32)	PD-A and PD-NA might involve different brain processes. At the same time, ALFF in the slow-5 band and fALFF in the regular band can tell PD-A and PD-NA apart.

PSA: post-stroke apathy; nPSA: non-post-stroke apathy; HCs: healthy controls; aAD: apathetic AD; naAD: nonapathetic AD; CN: control normal; PD-A: PD patients with apathy; nPD-A: PD patients without apathy; ALFF: amplitude of low-frequency fluctuation; fALFF: fractional amplitude of low-frequency fluctuation.

## Data Availability

The data that support the findings of this study are available from the corresponding author upon reasonable request. The study is still ongoing, and other variables related to apathy are being collected and analyzed. These variables will be reported in future publications.

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
