# Peer review of "Altered Resting-State Brain Activity and Functional Connectivity in Post-Stroke Apathy: An fMRI Study"

_brainsci, 2023, doi:10.3390/brainsci13050730_

Round 1
Reviewer 1 Report
The manuscript should be edited to improve presentation. The study is interesting, but some other works are related to the aim of this study.
The following articles should be considered to enrich introduction and discussion: Ferro, J. M., Caeiro, L., & Figueira, M. L. (2016). Neuropsychiatric sequelae of stroke. Nature Reviews Neurology, 12(5), 269-280.
Douven, E., Köhler, S., Rodriguez, M. M., Staals, J., Verhey, F. R., & Aalten, P. (2017). Imaging markers of post-stroke depression and apathy: a systematic review and meta-analysis. Neuropsychology Review, 27, 202-219.
Sutoko, S., Atsumori, H., Obata, A., Funane, T., Kandori, A., Shimonaga, K., ... & Tsuji, T. (2020). Lesions in the right Rolandic operculum are associated with self-rating affective and apathetic depressive symptoms for post-stroke patients. Scientific reports, 10(1), 1-10.
Li, Y., Luo, H., Yu, Q., Yin, L., Li, K., Li, Y., & Fu, J. (2020). Cerebral functional manipulation of repetitive transcranial magnetic stimulation in cognitive impairment patients after stroke: an fMRI study. Frontiers in Neurology, 11, 977.
All labels in figures should be edited for clarity. Not just to improve resolution, but also the group from which the data were acquired (or the comparisson) should be clear in each figure.
Supplementary figure should be in other file. Is it ANOVA of PSA? What is the meaning of this figure?
Conclusions should be clear regarding the putative role of mentioned networks in apathy. As well as the differences with other related studies.
When authors declare the study is not completed (in data availabitily)...they mean other variables have being studied? This should be clear.
Author Response
Dear Reviewer,
Thank you for giving us the opportunity to submit a revised draft of the manuscript “Altered resting-state brain activity and functional connectivity in post-stroke apathy: an fMRI study” for publication in the Brain Sciences. We appreciate the time and effort that editors and reviewers dedicated to providing suggestions on our manuscript, and are grateful for the insightful advice on and valuable improvements to our paper. We have incorporated all of the suggestions made by the reviewer. Those changes are highlighted within the manuscript. Please see below, in red, for a point-by-point response to the reviewer’ comments and concerns. All page numbers refer to the revised manuscript file with tracked changes.

Reviewer 2 Report
The authors present the article entitled “Altered resting-state brain activity and functional connectivity in post-stroke apathy: an fMRI study”
The purpose of this study is to investigate the changes in brain activity and functional connectivity in stroke patients with and without apathy.
The article presents the following concerns:
- Please use passive voice instead of using personal pronouns.
- Authors are asked to put quantitative results of their results in the abstract.
- At the end of the introduction section, please add a description of the manuscript's structure.
- Please, in the introduction, add the main contributions of the work, as well as the controversy to be resolved, the proposed hypothesis, and the objectives of the investigation since they need to be more clearly and explicitly stated.
- Add a brief introduction between section 2 and subsection 2.1.
- Please mention the acceptance and exclusion criteria for healthy volunteers.
- The authors mention that if image quality problems occurred during the data preprocessing phase, the subjects were also excluded. Therefore, they must define what were the quality problems considered and if there were a few subjects that were excluded.
- Add a brief introduction between section 3 and subsection 3.1.
- The format of table 1 should be improved since it needs to be clarified from the columns.
- ANOVA is an excellent statistical method to compare the PSA and nPSA groups. I recommend the following references with similar biomedical works to give more examples in this extent in line “...roimaging variables were compared using one-way ANOVA among the…” A novel methodology for classifying emg movements based on svm and genetic algorithms; Differences in the visual performances of patients with strabismus, amblyopia, and healthy controls; Impact of eeg parameters detecting dementia diseases: a systematic review; A novel method for measuring subtle alterations in pupil size in children with congenital strabismus.
- Please, decrease the % of similitude since the manuscript has 48%, according to Turnitin. It would be well less than 20%.
- Correct the caption of figure 1.
- Please move Figure S1 right after it was mentioned.
- The quality of the Figures must be improved since there are some of them, like figure four, that text cannot be distinguished.
- Figure 1 must be analyzed in more detail.
- Please define what the X, Y, and Z variables represent, what units they have, and what the T score represents.
- Please detail and explain figure 3 in-depth, as it contains multiple subfigures that are not mentioned.
- The style of citation is different from the journal’s style.
- The authors mention that "our study was the first to elaborate the relationship between PSA and spontaneous brain activity, as well as the functional connectivity of important nodes involved in large-scale intrinsic connectivity networks (ICNs), such as the default mode network." (DMN). , central executive network (CEN), salience network (SN), and limbic system." However, it needs to be clarified in the text where the functional connectivity analysis of important nodes involved in large-scale intrinsic connectivity networks (ICNs), such as the default mode network (DMN), is performed. Please clarify and complement the information mentioned in the Materials, Methods, and Results sections.
- The authors must place in the discussion section a table with the main achievements and results where they are compared with similar works.
- Update references since there are only 12 references of 60 older than 2018.
The following misspelling should be checked:
- Use words instead of numbers when they are less than three digitspage 4: “Prior to statistical analysis…” should be rewritten as “Before statistical analysis…”
- The abbreviation “vs” seems to be incorrectly punctuated. Changing by “vs.”
- page 8: “the first to elaborate the relationship…” should be rewritten as “elaborate on the relationship…”
- page 8: “in a resting state…” may be redundant. Consider changing by “resting”
Author Response

(The authors gave the same response as above.)

Reviewer 3 Report
The manuscript deals with an important problem namely the reduced quality of life after the stroke. The authors aimed at investigation of changes in brain activity in stroke patients with and without apathy. The cohort consisted of 21, 37 and 29 subjects with PSA, without PSA and healthy, respectively. Apathy was assessed at three months after stroke using the apathy diagnosis and Apathy Evaluation Scale (AES). Fractional amplitude of low frequency fluctuation (fALFF) was used to measure spontaneous brain activity, and region-of-interest to region-of-interest analysis to examine functional connectivity. The paper is well written and easy to follow. Methods used are adequately described. The conclusions are supported by the results. The fALFF values in the left middle temporal gyrus, right anterior and middle cingulate gyrus, right middle frontal gyrus, and right cuneus gyrus differed significantly (P<0.05) among groups and were positively correlated with apathy severity. The functional connectivity analysis revealed that reduced connectivity between the frontal lobe, temporal lobe, cingulate gyrus, and cuneus was associated with PSA. The authors stated that larger cohorts from different hospitals are required to confirm the findings.
The paper needs only minor revision.
FOV = 200 × 200 mm. After functional -> FOV = 200 × 200 mm). After functional
HAMD score is mentioned both as significantly different and not in Tab.1
Author Response

(The authors gave the same response as above.)

Round 2
Reviewer 2 Report
The manuscript can be accepted